# Mesenchymal stem cell-derived exosome and liposome hybrids as transfection nanocarriers of Cas9-GFP plasmid to HEK293T cells

Behnaz Gharehchelou[1], Mehrnoush Mehrarya[1], Yahya Sefidbakht[1,2]*, Vuk Uskoković[3,4]*, Fatemeh Suri[5], Sareh Arjmand[1,2], Farnaz Maghami[1], Seyed Omid Ranaei Siadat[1], Saeed Karima[6], Massoud Vosough[7]*

**1** Protein Research Center, Shahid Beheshti University, Tehran, Iran, **2** Department of Cell and Molecular Biology, Faculty of Life Sciences and Biotechnology, Shahid Beheshti University, Tehran, Iran, **3** TardigradeNano, LLC, Irvine, CA, United States of America, **4** Division of Natural Sciences, Fullerton College, Fullerton, CA, United States of America, **5** Ophthalmic Research Center, Research Institute for Ophthalmology and Vision Science, Shahid Beheshti University of Medical Sciences, Tehran, Iran, **6** Department of Clinical Biochemistry, Shahid Beheshti University of Medical Sciences, Tehran, Iran, **7** Department of Regenerative Medicine, Cell Science Research Center, Royan Institute for Stem Cell Biology and Technology, ACECR, Tehran, Iran

* y_sefidbakht@sbu.ac.ir (YS); vuskokovic@fullcoll.edu (VU); masvos@royaninstitute.org (MV)

**Data Availability Statement:** All Data are included in the manuscript and supplementary the uncroped SDS-gel image is also attached in "S1 raw images".

## Abstract

Exosomes are natural membrane-enclosed nanovesicles (30–150 nm) involved in cell-cell communication. Recently, they have garnered considerable interest as nanocarriers for the controlled transfer of therapeutic agents to cells. Here, exosomes were derived from bone marrow mesenchymal stem cells using three different isolation methods. Relative to filtration and spin column condensation, the size exclusion chromatography led to the isolation of exosomes with the highest purity. These exosomes were then hybridized with liposomes using freeze-thaw cycles and direct mixing techniques to evaluate whether this combination enhances the transfection efficiency of large plasmids. The efficiency of these hybrids in transferring the Cas9-green fluorescent protein plasmid (pCas9-GFP) into the human embryonic kidney 293T (HEK293T) cells was evaluated compared to the pure exosomes. Both Cas9-GFP-loaded exosomes and exosome-liposome hybrids were taken up well by the HEK293T cells and were able to transfect them with their plasmid loads. Meanwhile, the treatment of the cells with plasmids alone, without any vesicles, resulted in no transfection, indicating that the exosome and exosome-liposome hybrids are essential for the transfer of the plasmids across the cell membrane. The pure exosomes and the hybrids incorporating liposomes obtained by the heating method transfected the cells more efficiently than those containing liposomes obtained by the thin film hydration technique. Interestingly, the method of combining exosomes with liposomes (freeze-thaw *vs.* direct mixing) proved to be more decisive in determining the size of the vesicular hybrid than their composition. In contrast, the liposome component in the hybrids proved to be decisive for determining the transfection efficiency.

**Funding:** The Iran National Science Foundation (INSF) grant No. 99025522 is gratefully acknowledged for support. The funders had no role in study design, data collection and analysis, decision to publish, or preparation of the manuscript.

**Competing interests:** no conflict of interest.

## 1. Introduction

By utilizing both natural and artificial nanovesicles, biomedical researchers have created efficient strategies for the treatment of disorders such as cancer in recent years [1, 2]. Extracellular vesicles (EVs) are important mediators of intracellular communication by transporting different cargos including mRNA, microRNA, metabolites, lipids, and proteins [3, 4]. EVs come in micro- and nano-sized forms and are categorized into three main types based on their size: apoptotic bodies, microvesicles, and exosomes [5, 6]. These categories are detailed in Table 1. Briefly, exosomes, the smallest EVs, are derived from multivesicular bodies (MVBs) and have a lipid-rich membrane abundant with tetraspanins and cholesterol, which facilitate the genetic material transfer and intercellular communication. Microvesicles are larger than exosomes and form through plasma membrane budding, mirroring the parent cell's lipid composition with moderate cholesterol levels, and are crucial for cell-to-cell signaling in processes such as inflammation and tissue repair [7]. Apoptotic bodies, the largest EVs, form during cellular apoptosis, encapsulating DNA, histones, and organelles within less stable membranes and playing a key role in clearing cellular debris and modulating immune responses [8].

Since exosomes originate from the endosomal compartment in most eukaryotic cells, they can be isolated from various biological fluids, including tears, breast milk, blood, urine, and saliva [9, 10]. Exosomes, secreted by diverse cell types [11], carry specific cargoes that can be localized on their surface, such as CD9, CD63, and CS81, and which serve as markers indicating their cellular origin [12–14].

According to the literature, mesenchymal stem cells (MSCs) produce a larger amount of EVs, including exosomes, compared to other cell types such as embryonic kidney cells or myoblasts [14]. MSC-derived exosomes have been shown to replicate many of the therapeutic benefits of MSCs, including anti-inflammatory, immunomodulatory, paracrine and regenerative effects [15]. Hence, these exosomes have been considered as a non-cell-based therapy replacement for MSC-based therapies [16].

Different techniques, including differential ultracentrifugation (the gold standard), density gradients, ultrafiltration, size exclusion chromatography (SEC), precipitation, and numerous commercial kits, have been used to separate exosomes from diverse biological fluids and

**Table 1. Characteristics and comparison of exosomes, microvesicles, and apoptotic bodies.**

| Extracellular Vesicles | Exosomes | Microvesicles | Apoptotic Bodies |
|---|---|---|---|
| Size | 30–150 nm | 100–1000 nm | >1 µm |
| Origin | Multivesicular bodies (MVBs) | Plasma membrane, through outward budding | Plasma membrane, during apoptosis |
| Biogenesis | Formed inside MVBs and released via fusion with the plasma membrane | Bud directly from the plasma membrane | Form during apoptosis as the cell disassembles |
| Markers/ Components | Tetraspanins (CD9, CD63, CD81), Alix, Tsg101, flotillin-1 | Selectins, Integrins | DNA, histones, cell organelles, nuclear fractions, Annexin V |
| Applications | Transfer of genetic material (e.g., mRNA, miRNA), diagnostics | Cell signaling, communication | Modulation of immune response |
| Functions | Intercellular communication, cargo delivery | Facilitate communication between cells | Removal of apoptotic debris, immune signaling |
| Phospholipid Content | High in sphingomyelin and phosphatidylserine | Similar to the parent cell membrane | Diverse phospholipid composition reflecting cellular disassembly |
| Cholesterol Content | High | Moderate | Variable, often lower than exosomes and microvesicles |
| Membrane Stability | High stability due to lipid composition | Moderate stability | Generally less stable due to disintegration process |

clinical samples [4, 9, 17]. To enhance the effectiveness and purity of exosome separation, these strategies can be used individually or in combination with others [10]. Numerous studies have shown that SEC, often referred to as gel filtration chromatography, is a quick and effective way to produce pure EVs while maintaining vesicular architecture and function [7, 8]. SEC typically comprises two phases, a porous stationary phase, and a mobile phase, with the separation depending on the hydrodynamic volumes of the particles or differential molecular sizes [10, 18]. Sepharose (CL-2B and CL-4B) and Sephacryl (S-400) are examples of commercial columns used as the stationary phase, and phosphate-buffered saline (PBS) is the typical mobile phase [19, 20]. Before injecting the diluted samples, like those from the cell culture, into the chromatographic column, they need to be concentrated and rid of excess impurities. This can be achieved through various methods such as filtration-based techniques, centrifugation, precipitation, and size-exclusion chromatograph [21].

Exosomes are now regarded as natural nanostructured delivery systems with a number of advantages for their use as medication delivery agents. Suitable size, high biocompatibility, high stability, high loading efficiency for both hydrophilic and hydrophobic compounds, low immunogenicity, low cytotoxicity, a long half-life in circulation, the ability to safeguard cargo against deterioration, and the capacity to cross the blood-brain and placental barriers are some of these properties [6, 11, 12, 22, 23]. Despite these advantages, the small size of exosomes can limit their ability to transport large nucleic acid molecules, such as relatively large plasmids used in genetic engineering [24, 25]. To overcome this limitation, exosomes have been combined with synthetic nanocarriers like liposomes [25]. Another goal of merging liposomes with exosomes is to enhance cellular delivery by creating a hybrid system that can carry both lipophilic and hydrophilic compounds while maintaining the inherent biological properties of exosomes. Utilizing the benefits of liposomes can enhance the therapeutic potential of exosomes and facilitate the development of advanced drug delivery systems [26].

Liposomes are laboratory engineered, nanoscale vesicles with lipid bilayers that encapsulate substances, particularly those with poor bioavailability, making them valuable in drug delivery, diagnostics, and other biomedical applications [27]. The synthesis of liposomes can be achieved through several techniques, each tailored to produce vesicles with specific characteristics. For instance, mechanical methods, such as sonication and extrusion, physically manipulate the lipid components to form liposomes [28]. Solvent dispersion methods involve the use of organic solvents to disperse lipids, which subsequently form bilayers upon solvent removal. Other methods involve vesicle fusion or size transformation to achieve desired structures. These approaches enable precise control over liposome properties such as size, lamellarity, and encapsulation efficiency for different scientific and therapeutic applications [29].

This study focuses on the synthesis of liposomes using two primary methods: a modified heating method and the thin film hydration technique, the latter of which is a subtype of the solvent dispersion method. These methods incorporate non-toxic components such as soybean phosphatidylcholine, cholesterol, glycerol, and vitamin E to enhance stability and biocompatibility. Exosomes were isolated from bone marrow-derived human mesenchymal stem cells (BM-hMSCs) through filtration, spin column condensation, and size exclusion chromatography, followed by characterization using dynamic light scattering, flow cytometry, and protein assays. The exosome-containing fractions were then combined with liposomes using freeze-thaw cycles and direct mixing techniques. The Cas9-green fluorescent protein plasmid was used to evaluate the gene delivery efficacy of these liposome-exosome hybrids in human embryonic kidney 293T cells. The aim of this study is to assess the transfection efficiency and potential of these hybrids as gene delivery vectors.

## 2. Materials and methods

### 2.1. Cell culture

The HEK293T cell line was purchased from the Iranian Biological Resource Center. The HEK293T cells were cultured in Dulbecco's Modified Eagles Medium/Nutrient Mixture F-12 (DMEM F12) containing 10% Fetal Bovine Serum (FBS) and 1% penicillin-streptomycin (all from Bioidea Co., Tehran, Iran). BM-hMSCs were purchased from the Royan Institute, Tehran, Iran, and cultured in minimum essential alpha (MEM-α) medium supplemented with 5% human platelet lysate (hPL) at 37˚C and in the presence of 5% $CO_2$. The cells were passaged at 75–80% confluency. The cell culture was conducted in a GMP-compliant clean room to maintain the quality and safety of the exosome products.

### 2.2. Exosome isolation

The exosomes were isolated from the BM-hMSCs supernatant (obtained from the Royan Institute, Tehran, Iran) using three different methods, including SEC, EXOCIB exosome isolation kit (Cibbiotech, Tehran, Iran), and precipitation with polyethylene glycol 4000 (PEG4000) (Merck, Darmstadt, Germany). All isolated exosomes were stored at -20˚C for the next evaluations [30]. To achieve this, 20 ml of the conditioned media was obtained from 8–10 million cells in a 150 $cm^2$ flask following a three-day incubation period. The purification process was carried out in a cold room. All fractions were filtered using a 0.22 μm filter under a laminar flow hood to prevent contamination and ensure sterility.

**2.2.1. SEC method.** For the isolation process using SEC, a column was designed and packed with Sepharose CL-4B resin (supplied by Noavaran Zist Gostar ARG Company, Tabriz, Iran). The column had a diameter of 1.6 cm, a height of 16 cm, and a volume of approximately 32 $cm^3$. The column was washed and equilibrated with PBS. Fifty milliliters of the BM-hMSCs supernatant was centrifuged ($500 \times g$ at 4˚C for 5 mins and then at $3000 \times g$ at 4˚C for 10 mins) to remove dead cells and cell debris. The resulting supernatant was filtered through a 0.22 μm filter and concentrated 40-fold using a 100 kDa filter (Corning Spin-X UF centrifugal concentrator, 100000 MWCO Membrane) through repeated centrifugation at $3000 \times g$ at 4˚C for 25 mins. The concentrated supernatant (1 ml) was centrifuged at $13000 \times g$ for 30 min before being applied to the SEC column at a 0.9 ml/min flow rate. A total of 40 fractions, each 1 ml, were collected from the column. The fractions containing exosomes were identified by flow cytometry (fractions 6–10) and were separated and concentrated for further applications.

**2.2.2. EXOCIB kit.** To isolate the exosome using the EXOCIB kit, the BM-hMSCs supernatant was centrifuged twice and filtered through a 0.22 μm filter, as explained in 2.2.1. The filtered supernatant was applied for exosome isolation according to the kit manufacturer's instructions [31]. Briefly, the filtered media were incubated with the precipitation solution overnight (12 h) at 4˚C. After incubation, the mixture was centrifuged at 3000 rpm for 40 min, and the resulting pellet was collected. The isolated exosome pellet (E-exosome) was subsequently dissolved in 1 ml of PBS.

**2.2.3. Precipitation with PEG4000.** Polyethylene glycol (PEG)-based precipitation is widely recognized as an efficient and scalable method for isolating extracellular vesicles, including exosomes, due to its capacity to concentrate vesicles from large volumes of supernatants [32]. Here, the BM-hMSCs supernatant was centrifuged twice and filtered through a 0.22 μm filter, as explained in 2.2.1. The filtered supernatant was mixed with the PEG4000 solution (5% PEG in 1.5 M NaCl) and subjected to a 1 h incubation period with the sonication interval that was set at 20 s ON, 120 s OFF. After incubating overnight at 4˚C, the solution was centrifuged at $3000 \times g$ for 40 mins. The resulting exosome pellet (P-exosome) was dissolved in 1 ml PBS.

## 2.3. Exosomes characterization

Several techniques were applied to prove that the isolated fractions contained exosomes, including; I) scanning electron microscopy (SEM), Transmission electron microscopy (TEM), and dynamic light scattering (DLS) to assess the exosomes' size and/or morphology, II) flow cytometry to detect exosomes markers, and III) Bradford assay and SDS-PAGE electrophoresis to detect the protein concentration of SEC fractions and exosomes.

**2.3.1. Electron microscopy.** The SEM (MIRA3 TESCAN, Czech Republic) was employed to characterize the isolated exosomes in terms of their dimensions, morphology, and structural integrity. For SEM sample preparation, purified exosomes from SEC the method (fractions 6–10, which met the criteria for purified exosomes, were combined as a single sample) were initially fixed in a 2.5% glutaraldehyde solution in PBS for 10 min. Following fixation, the exosomes were sputter-coated with a thin layer of gold to enhance electron signal conductivity. Finally, SEM imaging was performed. TEM imaging was conducted using a Philips CM120 microscope (Netherlands). To prepare the specimens for TEM, the exosome aliquot was dispersed onto a 300 mesh copper formvar/carbon grid and stained with uranyl acetate. The TEM was operated at a voltage of 80 kV.

**2.3.2. Dynamic light scattering (DLS).** The hydrodynamic diameter distribution of the fractions was analyzed using DLS with the Cordouan Zetasizer (Cordouan Technologies, France) at room temperature (RT). To achieve optimal measurement conditions and avoid multiple scattering effects, the purified exosomes were diluted 1:2 with filtered (0.22 μm) PBS [33]. The data were processed and interpreted using NanoQ software (version 2.5.9.0), which provided a detailed insight into the size and distribution of the exosomes.

**2.3.3. Flow cytometry.** Flow cytometry was employed to identify membrane-bound proteins, specifically the classic surface markers of exosomes, CD63, CD9, and CD81, as previously described [34, 35]. For this analysis, 150 μl of the isolated exosome sample in PBS was incubated with aldehyde/sulfate-latex beads for 15 min at RT. The beads were then resuspended in bead-coupling buffer (BCB) and incubated at RT on rotation for 2 h. The exosome-coated beads were subsequently labeled separately with monoclonal antibodies against human CD63, CD9, and CD81 (eBioscience, San Diego, CA) at 4°C for 30 mins. After labeling, the beads were washed with BCB, and analyzed by flow cytometry using a FACS Calibur cytometer (BD, New San Jose, CA). The data were processed using WinMDI 2.8 software.

**2.3.4. Bradford assay and SDS-PAGE electrophoresis.** The total protein content of all fractions was assessed using a Bradford protein assay [36]. The fraction samples were incubated at RT for 45 min before the assay. The measured absorbances at 595 nm were quantified relative to a standard curve prepared by serial dilutions of bovine serum albumin (BSA) proteins.

The samples were investigated for expression of exosome markers (CD63, CD9) and protein content using SDS-PAGE. Collected fractions from the SEC method, the samples precipitated using the extraction kit and PEG4000 (40 fractions, E-exosomes, and P-exosomes) were denatured in Laemmli buffer and boiled at 95°C for 5 min to reduce disulfide bonds in proteins. Subsequently, 0.01 ml of each sample was loaded into each well of a 10% polyacrylamide gel. Electrophoresis was performed at a constant voltage of 100 V for 120 mins. Finally, the proteins in the resolving gel were stained with Coomassie Brilliant Blue G-250 dye.

## 2.4. Synthesis of liposomes and pCas9-GFP loaded liposome

Two types of liposomes were synthesized: liposome A and liposome B. Liposome A was prepared using a modified version of the heating method, also known as the Mozafari method [37]. The liposomal formulation consisted of soybean phosphatidylcholine (PC), cholesterol,

glycerol, and vitamin E. Vitamin E was included as a stable component that provides antioxidant properties, thereby protecting biological membranes and ensuring stability [38]. PC and cholesterol were hydrated in PBS at RT under a nitrogen atmosphere for 60 min. In the subsequent step, the cholesterol was heated to 120˚C, and after adding glycerol, it was heated for an additional 15 min until the cholesterol was completely dissolved. The temperature was then reduced to 60˚C, and PC and vitamin E were added to the glycerol solution. The pCas9-GFP plasmid (15 μg) and calcium ions (Ca$^{2+}$, 50 mM) were incorporated into the liposomes (285 μg) as the temperature and stirring speed were reduced to 45˚C and 100 rpm, respectively. After 15 min, the temperature and stirring were stopped, and the formed liposomes were kept at RT under a neutral gas (nitrogen) for 15 min. Finally, the liposomes were stored at 4˚C.

Liposome B was generated using the thin film hydration method [23, 39]. The lipid and polymeric components, consisting of dipalmitoylphosphatidylcholine (DPPC), cholesterol, and PEG4000 at a molar ratio of 88:10:2, were dissolved in chloroform in a round-bottom flask. The organic solvent was removed using a rotary evaporator (IKA, Deutschland, Germany) at 50˚C for 30 mins, resulting in the formation of a thin film. To prepare the pCas9-GFP-loaded liposome, the thin film was suspended and hydrated with PBS containing the pCas9-GFP plasmid at 60˚C, followed by gentle stirring for 30 min. The formed liposomes were then homogenized using a probe sonicator for 60 min, with a sonication cycle set to 10 s ON and 240 s OFF. This process was conducted in a cooled room to ensure efficient sonication and prevent DNA degradation. Subsequently, the liposomes were incubated at 4˚C to restore membrane integrity and prevent aggregation. To verify the integrity of the plasmid DNA, agarose gel electrophoresis was performed. The genetic map of the pCas9-GFP plasmid is shown in Fig 3C.

## 2.5. Preparation of exosome-liposome hybrids and exosomes-pCAS9-GFP complexes

Exosome-liposome hybrids were prepared using two techniques: freeze-thaw and direct mixing [22, 23, 40]. For the freeze-thaw method, exosome and plasmid-liposome complexes were mixed in a 1:1 volume ratio, incorporating 0.2% PEG4000. This mixture was placed in a nitrogen tank at -196˚C for one second, followed by a 20 min incubation at RT. These steps were repeated four times to enhance hybrid formation. In the direct mixing method, exosome and plasmid-liposome complexes were mixed in a 1:1 volume ratio and incubated at 37˚C for 12 h. Moreover, exosomes and pCAS9-GFP were mixed in a 2:1 volume ratio using the same direct mixing method.

## 2.6. Replication and isolation of pCas9-GFP

The pCas9-GFP was amplified in the *Escherichia coli* (DH5α) host cells. Overnight bacterial cultures were grown in the presence of ampicillin (100 μg/ml) as an antibiotic. The alkaline lysis method was employed to isolate the plasmid from these cultures. To further purify the isolated plasmids, silica gel column chromatography (Yekta Tajhiz Azma, Iran) was utilized. The extracted plasmids were then stored at −20˚C until needed for subsequent experiments or applications.

## 2.7. Transfection of HEK293T cells with pCas9-GFP

To evaluate the ability of exosomes (fractions 6–10 obtained from the SEC method) and exosome-liposome hybrids to serve as transfection agents for delivering pCas9-GFP into HEK293T cells, we prepared five distinct samples, as detailed in Table 2, along with positive and negative controls.

Polyethyleneimine (PEI MAX, MW 40,000) and liposome A were utilized as positive control transfection agents. HEK293T cells were seeded at a concentration of $0.5 \times 10^6$ cells per

**Table 2. Samples used in transfection experiments.**

| Samples | Components | Method |
|---|---|---|
| **1** | Exosome + liposome A+ 0.2% PEG4000 | Freeze-thaw |
| **2** | Exosome + liposome A | Direct mixing |
| **3** | Exosome + pCas9-GFP-loaded liposome B+ 0.2% PEG4000 | Freeze-thaw |
| **4** | Exosome + pCas9-GFP-loaded liposome B | Direct mixing |
| **5** | Exosome + pCas9-GFP | Direct mixing |
| **Positive Controls** | pCas9-GFP + culture medium without phenol red+ (PEI or liposome A) | Agitating for 15 mins |
| **Negative Control** | pCas9-GFP | - |

well in a six-well plate. Once the cells reached approximately 50% confluency, equating to about $1.2 \times 10^6$ cells per well, the culture media were replaced with fresh antibiotic-free media (DMEM-F12 + 10% FBS). Each sample was prepared with equal concatenations, containing pCas9-GFP and PEI at 1:3 ratio. This ensured that the final concentration of PEI in each well was maintained at 1 µg/µl (refer to Table 2).

After a 24 h incubation period, the transfected cells were examined using a fluorescence microscope equipped with a live-cell imaging system (Cytation 5; Biotek, Winooski, VT). The fluorescence emitted by the transfected cells was quantified using the ImageJ software package (NIH, Bethesda, MD) and compared to the control groups. To ensure accurate measurements across different samples, the quantified fluorescence was normalized to the cell monolayer surface area of 2,400 µm$^2$. The average fluorescence from four sample replicates was plotted for various samples and different amounts of exosomes using the OriginPro 8.5 software package.

## 3. Results

### 3.1. Isolation of BM-hMSCs-derived exosomes and analysis of protein impurities

Exosomes were separated from BM-hMSCs using the precipitation method (P-exosomes), EXOCIB exosome isolation kit (E-exosomes), and the sepharose CL-4B column in SEC. The 40 fractions of exosomes isolated on the packed sepharose column are shown in Fig 1A. The fractions were checked for the presence of exosome-specific biomarkers using flow cytometry (Section 3.4). According to the Bradford assay, the first 10 fractions were almost free of protein, while the highest protein contents were detected in fractions 15–28 (the highest for fraction 20 at 2.9 mg/ml) and P-exosomes and E-exosomes (3 and 3.3 mg/ml, respectively) (Fig 1B). The results of SDS-PAGE analyses supported this outcome (Fig 1C).

### 3.2. Electron microscopy

SEM micrographs of isolated exosomes (fraction 6–10, as explained in Sec.2.3.1) revealed their spherical form. According to the acquired images, the vesicle sizes range from 30 to just over 80 nm (Fig 2A and 2B). Concordantly, TEM micrographs depict exosomes with a well-defined spherical shape and visible membrane structures (Fig 2C and 2D). The lower exosome size observed in the TEM images as compared to SEM is due to the effects of the sample preparation, primarily gradient dehydration employed prior to TEM analyses [41].

### 3.3. DLS

The mean hydrodynamic diameters of purified exosomes, liposomes, and exosome-liposome hybrids (prepared by freeze/thaw or direct mixing) are shown in Table 3 and S1 Fig. The

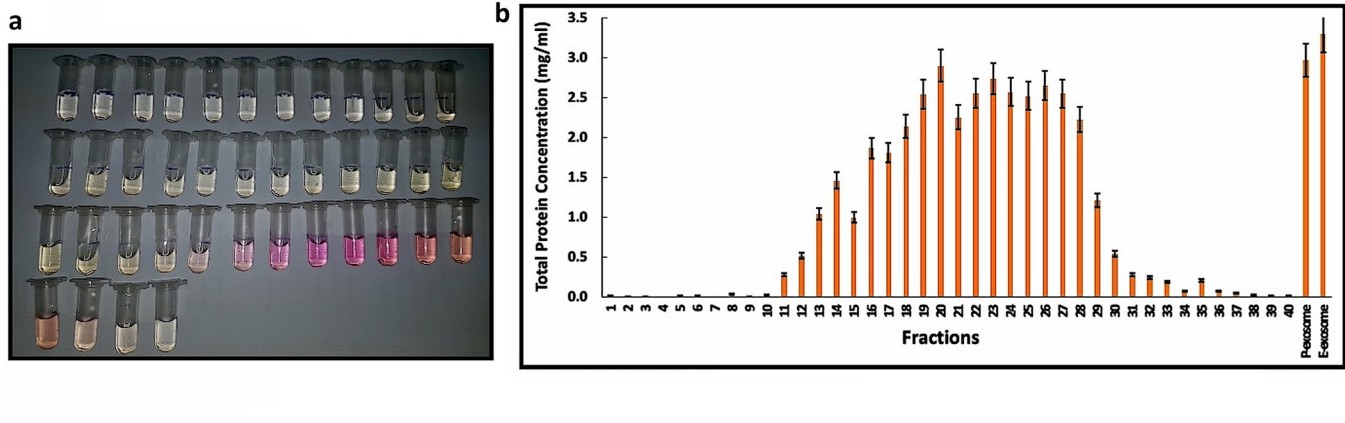

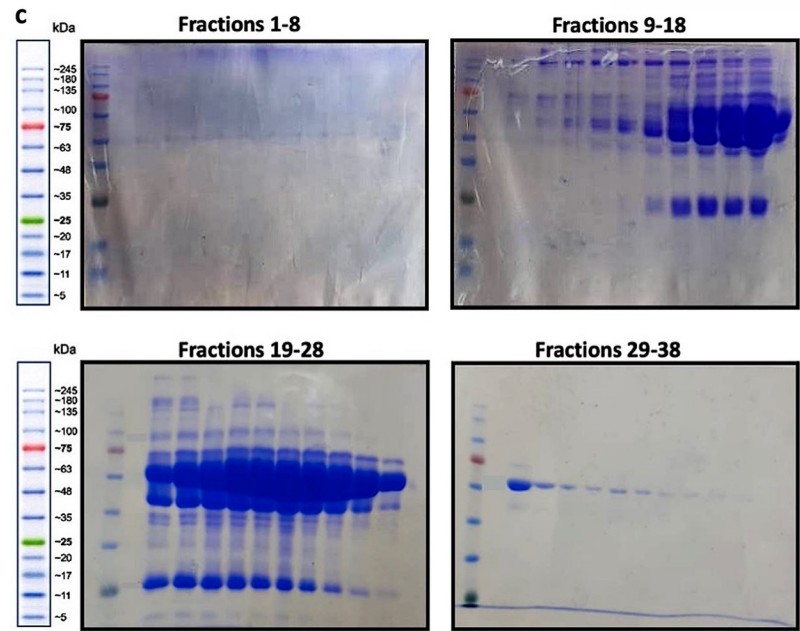

**Fig 1.** a) The 40 diluted fractions (1 ml each) were obtained after subjecting BM-MSCs to sepharose CL-4B column. b) Determination of exosome protein impurity concentrations in P-exosomes, E-exosomes, and 40 fractions isolated by SEC. The initial and the final fractions in samples isolated by SEC have the lowest amount of protein contamination, while the middle fractions and P- and E-exosomes have considerable protein impurities. c) The SDS-PAGE results for SEC fractions 1–38 show that the middle fractions contain a significantly higher concentration of protein, while the beginning and end fractions have few protein bands. This aligns with the Bradford assay findings.

increase in hydrodynamic diameters observed for exosome-liposome hybrids, as compared to the diameters of exosomes and liposomes alone, strongly suggests that fusion occurred between these two types of vesicles during the mixing process. Further, according to the obtained results, the hybrids obtained by the freeze/thaw methods were larger than those obtained by direct mixing. Therefore, it can be concluded that the method of preparation has a pronounced effect on the size of the exosome-liposome hybrids.

### 3.4. Flow cytometry

The exosome-specific marker, CD63, was detected by the flow cytometry in SEC-isolated fractions of both E-exosomes and P-exosomes, and the following results were obtained: 7.40% for fractions 6–10, 1.92% for fractions 11–15, 0.13% for fractions 16–20, 2.65% for fractions 21–25, 2.47% for E-exosomes, and 0.77% for P-exosomes (Fig 3). Two additional exosome-specific

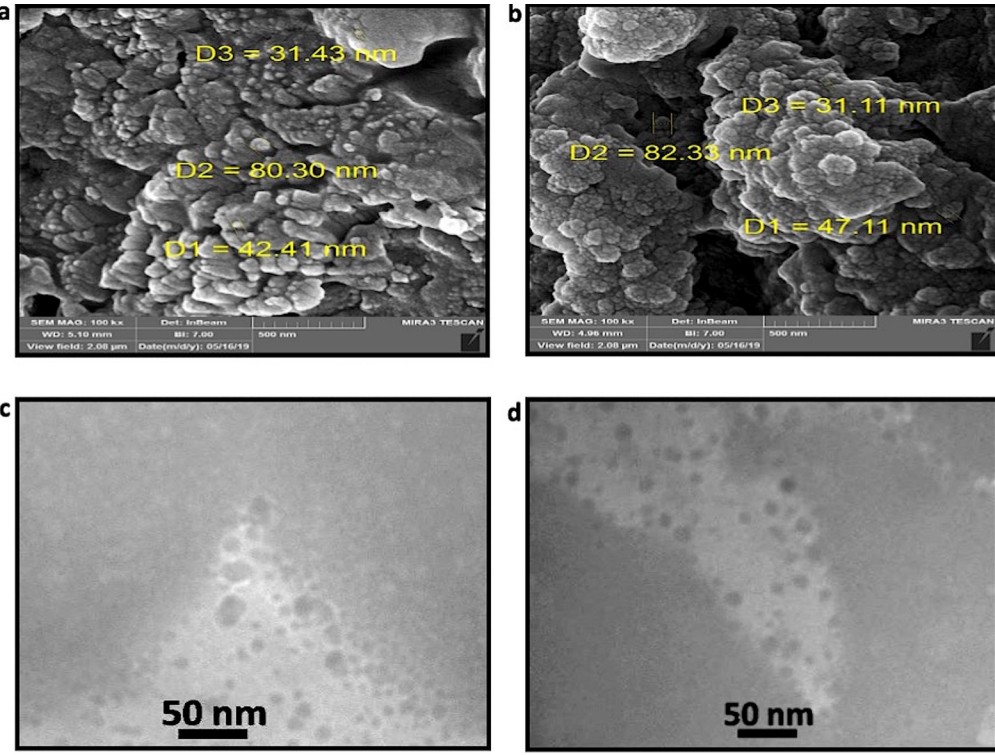

**Fig 2. Electron microscopy analysis of purified exosomes.** a and b) SEM images of isolated Exosomes reveal their individual size range of approximately 30 to 80 nm. c and d) TEM images reveal the existence of spherical and uniform vesicle shape characteristic of exosomes with the diameter of around 40 nm.

markers, CD90 and CD9, were detected in exosomes obtained from fractions 6–10, at the levels of 25.1% and 2.3%, respectively (S2 Fig).

## 3.5. Encapsulation efficiency analysis

The extracted plasmid structure is indicated in Fig 4C. The entrapment efficiency for the plasmid DNA measured by a NanoDrop spectrophotometer equaled 84.6%. This value is higher

**Table 3. DLS size measurement of different samples prepared in this study.**

| Samples | Mixing method | Hydrodynamic Diameter (nm) ± 20% |
|---|---|---|
| Exosome | - | 63.1 |
| Liposome A | - | 94.65 |
| Liposome B | - | 88.06 |
| Sample 1 (Exosome + liposome A+ 0.2% PEG4000) | Freeze-thaw | 540.63 |
| Sample 2 (Exosome + liposome A) | Direct mixing | 108.98 |
| Sample 3 (Exosome + pCas9-GFP-loaded liposome B+ 0.2% PEG4000) | Freeze-thaw | 827.21 |
| Sample 4 Exosome + pCas9-GFP-loaded liposome B | Direct mixing | 76.72 |

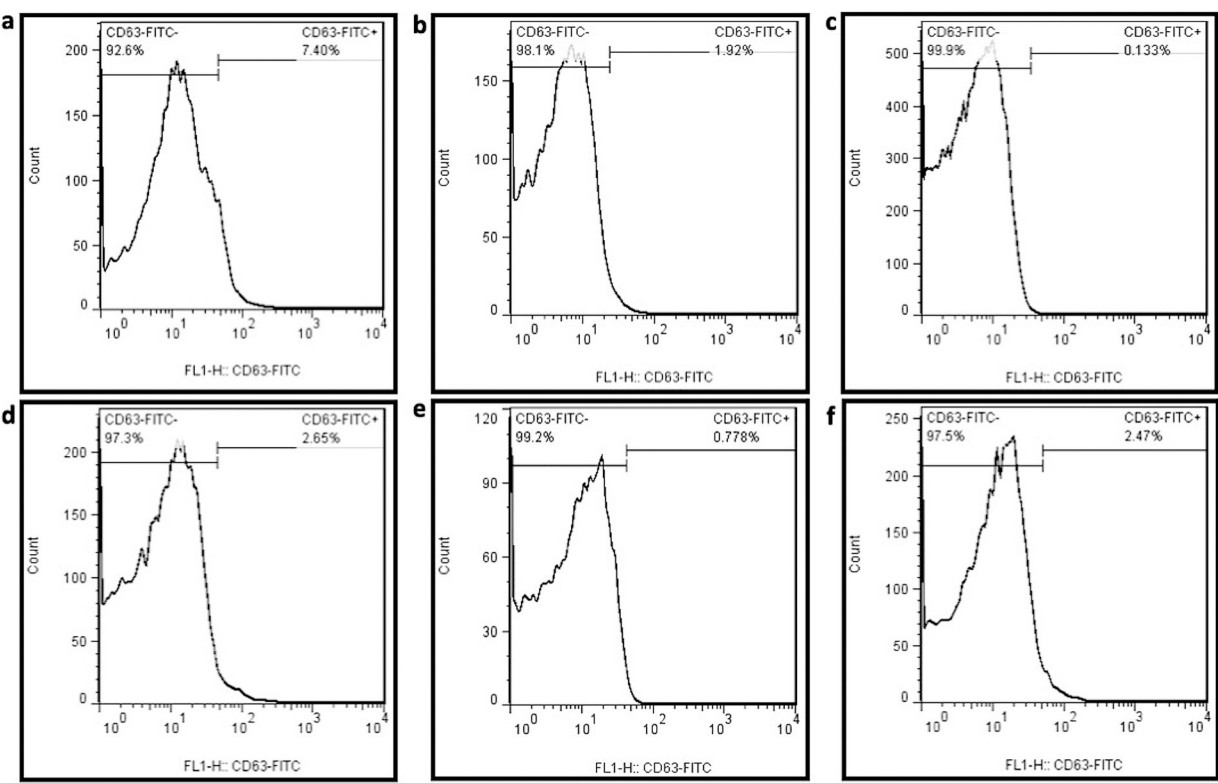

**Fig 3. BM-hMSCs-derived exosomes were characterized using flow cytometry.** The plots correspond to a) fraction 6–10, b) fraction 11–15, c) fraction 16–20, and d) fraction 21–25, e) P- exosomes, and f) E-exosomes.

than 70.3% plasmid entrapment for similar liposomes prepared by the original heating method [42].

### 3.6. Transfection of pCas9-GFP into the HEK293T cell line

Exosomes and exosome-liposome hybrids were both successful at delivering the pCas9-GFP plasmid to HEK293T cells, as shown in Fig 4. As expected, due to the absence of a transfection agent in the negative control (pCas9-GFP), the transfection process was not observed (Fig 4K), while using PEI and the liposome A in the positive control led to successful cell transfection (Fig 4I and 4J).

The fluorescence intensity comparison between cells exposed to various exosome-liposome samples reveals that the hybrids containing liposome A (Fig 4A and 4B) had a higher transfection efficiency than those containing liposome B (Fig 4C and 4D). Thus, the component of the liposomes in the exosome-liposome hybrids seems to be more decisive than the preparation method in determining the transfection efficiency. Interestingly, using exosomes alone as the transfection agent (sample 5) was also successful, with an even greater efficiency observed than for exosome-liposome hybrids (Fig 4A). Although the transfection efficiency results are comparable, any elaborate discussion of this difference is being hampered by the relatively large variability of the degree of transfection from cell to cell within a single cell specimen, which led to the statistical significance below the conventional levels of confidence ($p < 0.05$) for all hybrid sample group comparisons except that between the sample 3 hybrids and pure exosomes (sample 5). The key observation here, however, is that both pure exosomes and the exosome-liposome hybrids evidently increased the transfection efficiency relative to that achieved

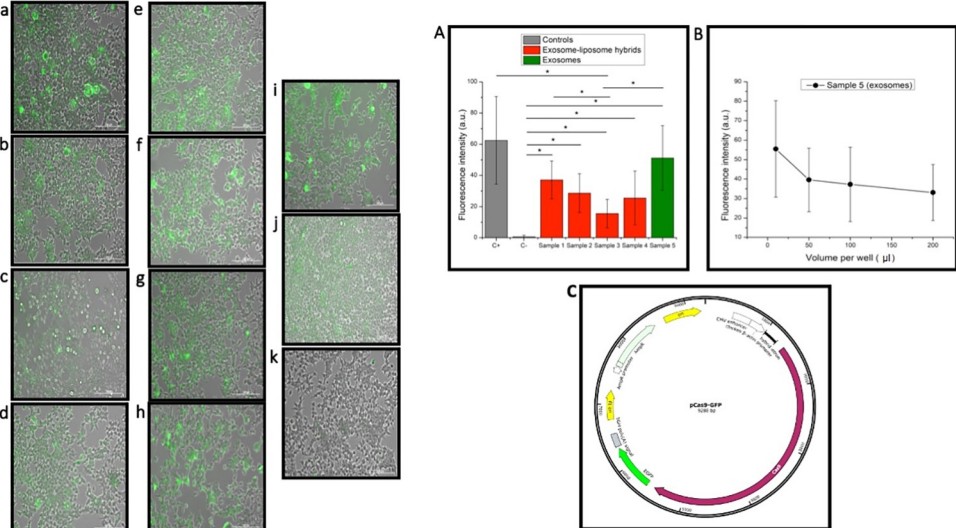

**Fig 4. Left Panel; Transfection of the HEK293T cell line with the Cas9-GFP plasmid.** a) 0.02 ml of sample 1 (Fractions 6–10 + liposome A+ 0.2% PEG4000). b) 0.02 ml of sample 2 (Fractions 6–10 + liposome A). c) 0.02 ml of sample 3 (Fractions 6–10 + pCas9-GFP-loaded liposome B+ 0.2% PEG4000). d) 0.02 ml of sample 4 (Fractions 6–10 + pCas9-GFP-loaded liposome B). e) 0.01 ml of sample 5 (Fractions 6–10 + pCas9-GFP). f) 0.05 ml of sample 5. g) 0.1 ml of sample 5. h) 0.2 ml of sample 5. i) Positive control (PEI). J) Positive control (liposome A). k) Negative control. Scale bar is 200 μm. right Panel; A) Comparison of the GFP fluorescence intensity, measured by ImageJ software, as the evidence of transfection in HEK293T cells treated with different transfection agents (samples 1–5) compared to the positive (C+) and the negative (C-) controls. Samples 1–4 are exosome-liposome hybrids (as explained in the text) and Sample 5 contains only exosomes as the transfection agent. * denote a statistically significant difference (p < 0.05). B) Transfection efficiency of increasing concentrations of exosomes when used as the transfection agent. Error bars represent standard deviation. C) The schematic representation of pCas9-GFP plasmid.

by the delivery of plasmids with no vesicular carriers (negative controls). On the other hand, increasing the exosome concentration negatively affected the transfection efficiency (Fig 4E–4H, and 4B) meaning that the optimization of exosome concentrations is of critical importance for a successful transfection to occur. In all, while the observed differences in transfection efficiency between exosomes and exosome-liposome hybrids were not statistically significant for all comparisons (except between sample 3 and pure exosomes), the data emphasize the variability and potential impact of vesicular carriers on transfection outcomes.

## 4. Discussion

In this study, MSCs were selected as the primary source of exosomes due to several compelling reasons. Notably, MSCs have a remarkable capacity to secrete a substantial amount of EVs, including exosomes. This characteristic makes them a highly efficient and dependable source for exosome isolation when compared to other cell types. The high yield of exosomes is especially beneficial for subsequent applications that require significant quantities of these vesicles [4, 15, 43].

Exosomes were isolated from BM-hMSCs culture supernatants using three different methods, namely SEC, EXOCIB exosome isolation kit, and precipitation with PEG4000. SEC was found to be the most advantageous technique, enabling easy and fast separation of exosomes with the highest purity and lowest protein contamination compared to the other two methods [10]. In the SEC purification method, a single column allows for the consecutive elution of differently sized extracellular vesicle samples. The separation and the recovery of vesicles are directly influenced by the height and the diameter of this column. For instance, the isolation of

proteins and EVs is improved by a column with a higher height and a smaller diameter [44]. In this study, sepharose CL-4B was chosen as the stationary phase due to its superior properties compared to other sepharoses, such as CL-2B and S-400. These properties include a wider range of pH stability, the ability to withstand denaturants without compromising the performance, and suitability for separating a diverse range of biomolecules based on their size and shape [45, 46]. Comparative studies show that the smaller intra-pores of the CL-4B resin ($\approx 40$ nm *vs.* $\approx 70$ nm for CL-2B) produce a stronger elution profile, which liberates exosomes from common protein contaminants like albumin and lipoproteins [19, 21, 45–47]. In a recent study, Guo et al, demonstrated that the exosome which were separated by CL-4B and CL-6B had higher purity and resolution [48]. Vesicles and proteins of varying sizes interact differently with the pores of the stationary phase, leading to their passage through the column at different speeds. Exosomes, having a larger hydrodynamic radius than smaller components, are unable to pass through the stationary phase pores. Consequently, they spend less time in the column, eluting earlier and appearing in the initial fractions, just after the column's void volume [21, 47, 49].

Theodoraki *et al.* demonstrated that conventional cytometers are effective at detecting and quantifying proteins in exosomes isolated from supernatants [50]. Initially, we assessed CD63 using flow cytometry, a member of the transmembrane-4 glycoprotein superfamily, known as tetraspanins. CD63 is particularly abundant in exosomal membranes and is widely recognized as a key biomarker for exosomes [51]. The flow cytometry results for CD63 markers in the collected fractions revealed a significantly higher presence in fractions 6–10 obtained through SEC. This finding confirms the superior isolation efficiency of SEC compared to the other two methods. To further evaluate these fractions, the expression of two additional exosome biomarkers, CD90 and CD9, was assessed [17]. The positive results for these biomarkers in the same fractions provide additional confirmation of their identity as exosomes.

While flow cytometry is a valuable method for identifying exosomes based on their surface markers, it does not guarantee that the sample is exclusively composed of exosomes, thus raising concerns about its purity [52]. Proteins are a significant impurity during exosome isolation [53]. Measuring the protein content of isolated exosomes revealed that samples obtained using the EXOCIB kit exhibited the highest protein concentration at 3.3 mg/ml, closely followed by those isolated through PEG4000 precipitation at 3 mg/ml. In contrast, the maximum protein content for exosomes isolated using SEC was lower, recorded at 2.9 mg/ml for fraction 20. This difference in protein yield may be due to the more effective removal of proteins during SEC, particularly in the initial fractions (6–10). Following the isolation and characterization of exosome fractions, the SEC-isolated exosomes (fractions 6–10) were chosen for subsequent experiments involving the electron microscopy, hybridization with liposome, encapsulation of plasmids and their subsequent transfer into HEK293T cells.

To validate the presence and examine the morphology of the exosomes, TEM was utilized. The TEM images clearly showed that the isolated particles possessed the characteristic spherical morphology associated with exosomes. The membrane structure of these vesicles was visible, confirming the successful isolation of intact exosomes. Furthermore, the consistent size and shape of the vesicles observed in the TEM images indicated a high level of purity in the exosome preparation, with minimal contamination from non-exosomal particles or debris. This morphological analysis substantiates the efficiency of the isolation protocol and is consistent with the flow cytometry data.

In this study, we utilized a Cas9 plasmid expressing the GFP protein as the model nucleic acid cargo. Previous research has demonstrated that exosomes can effectively encapsulate small nucleic acids and molecules, such as siRNA and doxorubicin; however, their efficiency in encapsulating larger entities is limited [54, 55]. To address this issue, we explored the

hybridization of exosomes with liposomes to determine whether this approach could enhance the transfection efficiency for a large cargo (Cas9-GFP plasmid). Despite differing greatly, there are several similarities between liposomes and exosomes, such as lipid bilayer, size, and their abilities to encapsulate therapeutic agents. These two systems may work well together by complementing each other's properties [9, 56]. In this study, we developed four different types of exosome-liposome hybrids for loading the large plasmid therein. Two types of liposomes (liposome A and liposome B) were chosen and hybrids with exosomes were produced using two different methods: direct mixing and freeze-thaw methods. The fusion of these two types of vesicles is facilitated by the lipid structure of both exosomes and liposomes.

Exosome-liposome hybrids prepared using the freeze-thaw method had a larger size than those prepared using the direct mixing method. However, since the small size of the nanocarriers is an advantage for gene delivery and gene therapies, as per the results of this study, the direct mixing method may be the method of choice for combining exosomes and liposomes. Despite our expectations, a comparative analysis of transfection efficiency of HEK293T cells with pure exosomes and the exosome-liposome hybrid formulations showed almost comparable results, however, the hybrids containing the liposome A were more efficient than those comprising liposome B. Therefore, the type of phospholipids appears to be more important than the method of mixing when it comes to defining the cellular uptake and transfection efficiency [57, 58]. This marginally negative effect of liposome B on the cellular uptake may be due to their effect on proteins involved in the cell-hybrid interaction, or the cytotoxic effect of DPPC (one of the liposome B ingredients) [22].

The concentration of exosomes or exosome-liposome carriers used had a significant impact on transfection efficiency. Although higher initial amounts of exosomes resulted in relatively better transfection, increasing the exosome concentration up to 4-fold did not lead to further enhancement. This suggests that the plasmid-to-exosome ratio should be investigated and optimized before proceeding with in vivo or clinical transfection studies.

In general, exosomes and exosome-liposome hybrides could be a promising candidate for *in vivo* and *in vitro* gene editing studies because commonly used viral vectors have limitations and lead to concerns including cytotoxicity, long-term expression, and immunogenic response when delivering nucleic acids [59]. Therefore, the results presented here may be a good starting point for innovations on the natural vehicle design for various forms of genetic engineering and genome editing.

## Conclusion

Exosomes of high purity were isolated from BM-hMSCs, especially using the SEC method. Following the isolation, they were hybridized with liposomes and evaluated for their cellular transfection efficiency. The pCas9-GFP-loaded exosomes and exosome-liposome hybrids both showed positive results in HEK293T transfection. However, the results also indicated that the mixing method is the crucial factor for determining the size of the vesicular hybrids. In contrast, the liposome component proved to be a more important determinant of the transfection efficiency than the method of synthesis. These results highlight the necessity for optimization of both the composition and preparation methods to create effective hybrid exosomal-liposomal carriers for use as transfection agents.

## Supporting information

**S1 Fig. DLS analysis of purified exosome samples.** a) 63.1 nm (fractions 6–10), b) 540.63 nm (sample 1), c) 108.98 nm (sample 2), d) 827.21 nm (sample 3), e) 76.72 nm (sample 4). (TIF)

**S2 Fig.** BM-hMSCs-Derived Exosome Fractions 1–9 Characterized for a) CD90, and b) CD9 Markers, Showing 25.1% CD90 and 2.3% CD9 Expression.
(TIF)

**S1 Raw images.**
(PDF)

## Acknowledgments

The authors would like to acknowledge Dr. Sina Mozaffari-Jovin from Mashhad University of Medical Sciences for assistance with this study.

## Author Contributions

**Conceptualization:** Yahya Sefidbakht, Massoud Vosough.

**Data curation:** Yahya Sefidbakht, Sareh Arjmand, Farnaz Maghami.

**Formal analysis:** Yahya Sefidbakht, Vuk Uskoković, Farnaz Maghami.

**Funding acquisition:** Yahya Sefidbakht.

**Investigation:** Yahya Sefidbakht.

**Methodology:** Yahya Sefidbakht, Massoud Vosough.

**Project administration:** Yahya Sefidbakht.

**Resources:** Yahya Sefidbakht, Seyed Omid Ranaei Siadat, Saeed Karima.

**Software:** Yahya Sefidbakht, Vuk Uskoković, Saeed Karima, Massoud Vosough.

**Supervision:** Yahya Sefidbakht.

**Validation:** Yahya Sefidbakht, Massoud Vosough.

**Visualization:** Yahya Sefidbakht, Vuk Uskoković.

**Writing – original draft:** Behnaz Gharehchelou, Mehrnoush Mehrarya, Yahya Sefidbakht, Vuk Uskoković, Sareh Arjmand, Massoud Vosough.

**Writing – review & editing:** Yahya Sefidbakht, Vuk Uskoković, Fatemeh Suri, Sareh Arjmand, Massoud Vosough.

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
