## [Decision Letter · Decision Letter 0]

9 Jun 2024

PONE-D-24-15406Mesenchymal Stem Cell-Derived Extracellular Vesicles and Liposome Hybrids as Transfection Nanocarriers of Cas9-GFP Plasmid to HEK293T CellsIf applicable, we recommend that you deposit your laboratory protocols in protocols.io to enhance the reproducibility of your results. Protocols.io assigns your protocol its own identifier (DOI) so that it can be cited independently in the future. For instructions see: https://journals.plos.org/plosone/s/submission-guidelines#loc-laboratory-protocols. Additionally, PLOS ONE offers an option for publishing peer-reviewed Lab Protocol articles, which describe protocols hosted on protocols.io. Read more information on sharing protocols at https://plos.org/protocols?utm_medium=editorial-email&utm_source=authorletters&utm_campaign=protocols.

We look forward to receiving your revised manuscript.

Kind regards,

Abdul Qadir Syed, PhD

Academic Editor

PLOS ONE

Journal Requirements:

3. Thank you for stating the following financial disclosure: "The Iran National Science Foundation (INSF) grant No. 99025522 is gratefully acknowledged for support."

4. In the online submission form, you indicated that available upon request.

8. PLOS ONE now requires that authors provide the original uncropped and unadjusted images underlying all blot or gel results reported in a submission’s figures or Supporting Information files. This policy and the journal’s other requirements for blot/gel reporting and figure preparation are described in detail at https://journals.plos.org/plosone/s/figures#loc-blot-and-gel-reporting-requirements and https://journals.plos.org/plosone/s/figures#loc-preparing-figures-from-image-files. When you submit your revised manuscript, please ensure that your figures adhere fully to these guidelines and provide the original underlying images for all blot or gel data reported in your submission. See the following link for instructions on providing the original image data: https://journals.plos.org/plosone/s/figures#loc-original-images-for-blots-and-gels.   

Reviewers' comments:

Reviewer's Responses to Questions

**Comments to the Author**

1. Is the manuscript technically sound, and do the data support the conclusions?

Reviewer #1: Yes

Reviewer #2: Partly

2. Has the statistical analysis been performed appropriately and rigorously? 

Reviewer #1: Yes

Reviewer #2: I Don't Know

3. Have the authors made all data underlying the findings in their manuscript fully available?

Reviewer #1: Yes

Reviewer #2: No

4. Is the manuscript presented in an intelligible fashion and written in standard English?

Reviewer #1: Yes

Reviewer #2: No

5. Review Comments to the Author

Reviewer #1: In the current manuscript, “Mesenchymal Stem Cell-Derived Extracellular Vesicles and Liposome Hybrids as Transfection Nanocarriers of Cas9-GFP Plasmid to HEK293T Cells ” the authors have shown the importance and The efficiency of these EV and Liposome hybrid system in delivering the genetic material into mammalian cells. They claim this hybrid system to be better and efficient than the conventional transfection methods. I recommend the manuscript for publication subject to responses to the following comments.

Minor comments

1) Authors should consider adding line numbers in the manuscript which would help in navigating and making comments.

2) Authors should consider increasing the resolution of the figures.

3) Why the authors have particularly choosen only fractions 6-10 and excluded fractions 1-5 and 37-40? What was the selection criteria?

4) The authors have used three different methods for EV isolation and have characterized SEC isolated EVs only by FACS while the EVs separated using the precipitation method and EXOCIB were further characterized by EM, DLS. What is the reasoning behind this criteria of selection. The author should consider explaining the results clearly.

5) The authors have used two methods for the preparation of EV-liposome hybrids. Previous studies suggest that Pegylation of Liposomes enhances the fusion of Liposomes with EVs. It would be interesting if authors could include an assay to compare the fusion efficiency of the two methods such as FRET based lipid mixing assay. There is a possibility of hemifusion between EV and Liposome, to compare the efficiency of the methods used to prepare EV-Lipo hybrids authors could also consider including Dithionite assay for accessing Hemifusion.

Reviewer #2: In this manuscript, Gharehchelou Xu et al., describe developing mesenchymal stem-cell derived extracellular vesicles and liposome hybrids as a transfection reagent for Cas9-GFP plasmids into HEK293T cells.

This study provides a promising start with preliminary data showing use of extracellular vesicle-liposome hybrids as a transfection reagent for a large plasmid such as Cas9-GFP in a single cell line-HEK293T. However, there are some concerns listed below including that the data is limited to a single cell type that needs to be addressed before this manuscript can be ready for publication.

Major points

1. The Introduction section should include background on the current state of transfection nanocarriers and the need for novel options. The Introduction is currently quite confusing and the authors tend to use extracellular vesicles and exosomes interchangeably. The Introduction would read better if these are properly explained in different sections, one for introduction of extracellular vesicles, categories (include a figure?), source, purification methods etc. Some background on liposomes and hybrid nanocarriers would also be useful as well.

2. The methods section needs a lot more detail.

- In Section 2.2: Were the BM-MSC patient-derived (human) or mouse?

- Section 2.2.1: How was supernatant concentrated down 40-fold? Was the total volume 40mls? This is not very clear from the written protocol.

- Section 2.2.2 and 2.2.3: What volume of PBS was the E- and P-Exosome pellets dissolved in?

- Section 2.3.1: Why was fraction 6-10 chosen to fix with 2.5 % glutaraldehyde. Please elaborate in the text.

- In Section 2.3.2, how was the EV diluted with PBS? What dilution factor was used?

- Section 2.4: What plasmid (pCas9-GFP) amount and Ca2+ amount was used? What is the neutral gas that was used here?

- It would be good to include the agarose gel electrophoresis data showing plasmid integrity as a supplementary figure.

- Section 2.7: Was a titration of the plasmid amount performed? Why was 15µg chosen? What concentration/amount of PEI was used?

- Please keep the time and volume units consistent throughout.

3. Results

- Please reference the figure number/sub-division while describing the result in the text, specifically in sections 3.1, 3.3 and 3.4

- Section 3.1 and Section 3.5 can be combined into one figure.

- Section 3.6: How was the trend analyzed? “Trend is evident” is a very vague statement.

- Was a titration performed with both variable EV and plasmid concentrations? If so, this data should be included.

- The authors are encouraged to test other transfection agents in comparison as a control? To compare and assess the effectiveness of using mesenchymal stem-cell derived EVs as a nanocarrier.

- The authors are also encouraged to demonstrate the effectiveness of the EV nanocarrier in other cell types including cancer cells, T/B cells or any primary cell type. HEK293T cells is one of the easiest to transfect cell type. So, it would be powerful to demonstrate the effectiveness of EVs in alternate cell types.

4. - Section 4: In the first line, the authors say that MSCs were used as the source of EVs as high amounts were extracted compared to other cell types. How was this determined? Were other cell types tested? If so, it would be great to include that data.

- What does ‘lowest protein contamination’ mean? How was this determined?

- What is the advantage of using CL-4B over other sepharoses? Describe and please provide a reference.

- In this manuscript, no titration was shown for EV or exosome-hybrids, to determine transfection efficiency. How can this be translation to in vivo or clinical studies if this information is not available in this manuscript.

5. Please edit the manuscript thoroughly.

6. PLOS authors have the option to publish the peer review history of their article (what does this mean?). If published, this will include your full peer review and any attached files.

Reviewer #1: No

Reviewer #2: No

---

## [Author Response · Author response to Decision Letter 0]

1 Oct 2024

PLOS ONE 17-Sep-24 

Dear Dr. Abdul Qadir Syed,

Thank you for handling our manuscript entitled “Mesenchymal Stem Cell-Derived Extracellular Vesicles and Liposome Hybrids as Transfection Nanocarriers of Cas9-GFP Plasmid to HEK293T Cells” to be considered for publication in the PLOS ONE. We appreciate all the comments from the reviewers that encouraged us to revise this manuscript. We have revised the manuscript thoroughly and addressed all the valuable comments from the reviewers. A detailed point-by-point response is appended and the changes made are highlighted in the text.

Thanks for your time and consideration.

Sincerely,

Yahya Sefidbakht; E-mail: y_sefidbakht@sbu.ac.ir

Reviewer #1: 

In the current manuscript, “Mesenchymal Stem Cell-Derived Extracellular Vesicles and Liposome Hybrids as Transfection Nanocarriers of Cas9-GFP Plasmid to HEK293T Cells ”Cells” the authors have shown the importance and The efficiency of these EV and Liposome hybrid system in delivering the genetic material into mammalian cells. They claim this hybrid system to be better and efficient than the conventional transfection methods. I recommend the manuscript for publication subject to responses to the following comments.

Minor comments:

1) Authors should consider adding line numbers in the manuscript which would help in navigating and making comments.

Response: Thanks for the comment. The line numbers were added to the text. 

2) Authors should consider increasing the resolution of the figures.

Response: Thank you for the comment. We have set all the figures to a resolution of 300 dpi in accordance with the journal’s guidelines.

3) Why the authors have particularly chosen only fractions 6-10 and excluded fractions 1-5 and 37-40? What was the selection criteria? 

Response: Thank you for this question. Due to the following criteria, we have chosen to analyze fractions 6-10 and excluded other fractions:

a) Purification: literature suggests that size exclusion chromatography can yield a purified exosome fraction(s). According to the Bradford assay results, the fractions 6-10 indicated this purity, with no protein contamination detected (Figure 1a). Furthermore, measuring OD280 confirmed the absence of protein contamination in these specific fractions. 

b) Exosome markers: flow cytometry data revealed the presence of CD90 and CD63, which are characteristic markers of exosomes, in these specific fractions (Figure 4 and S1). 

To address this comment, the above statements are now included in the manuscript, please see 

Section 2.2.1, line 165: “The fractions containing exosomes were identified by flow cytometry (fractions 6-10) and were separated and concentrated for further applications.”

4) The authors have used three different methods for EV isolation and have characterized SEC isolated EVs only by FACS while the EVs separated using the precipitation method and EXOCIB were further characterized by EM, DLS. What is the reasoning behind this criterion of selection. The author should consider explaining the results clearly.

Response: Thank you for this comment. When examining exosomes obtained through alternative techniques, referred to as P-exosomes and E-exosomes, we have found significant protein impurities, measuring 3 mg/ml and 3.3 mg/ml, respectively. Due to the mentioned protein contaminants, we decided not to utilize these products which were derived from these two methods (P-exosomes and E-exosomes). Thses subject were included in the revised version, page 6 section SEC, :

“The fractions containing exosomes were identified by flow cytometry (fractions 6-10) and were separated and concentrated for further applications.”

5) The authors have used two methods for the preparation of EV-liposome hybrids. Previous studies suggest that Pegylation of Liposomes enhances the fusion of Liposomes with EVs. It would be interesting if authors could include an assay to compare the fusion efficiency of the two methods such as FRET based lipid mixing assay. There is a possibility of hemifusion between EV and Liposome, to compare the efficiency of the methods used to prepare EV-Lipo hybrids authors could also consider including Dithionite assay for accessing Hemifusion.

Response: Thank you for your insightful suggestions. The primary objective of this study was to evaluate the transfection efficiency of exosomes and their liposome hybrids (mixture) prepared by non-toxic simple lipids, which limited the scope of our investigation into the fusion mechanisms. We agree that incorporating assays to compare the fusion efficiency of the two preparation methods, such as the FRET-based lipid mixing assay, would provide a deeper understanding of the interactions between exosomes and liposomes, however due to time and budgetary constraints, we are unable to include these additional analyses in the current study. We acknowledge the importance of these techniques and appreciate your recommendations. 

Reviewer #2: 

In this manuscript, Gharehchelou Xu et al., describe developing mesenchymal stem-cell derived extracellular vesicles and liposome hybrids as a transfection reagent for Cas9-GFP plasmids into HEK293T cells. This study provides a promising start with preliminary data showing use of extracellular vesicle-liposome hybrids as a transfection reagent for a large plasmid such as Cas9-GFP in a single cell line-HEK293T. However, there are some concerns listed below including that the data is limited to a single cell type that needs to be addressed before this manuscript can be ready for publication.

Major points

1. The Introduction section should include background on the current state of transfection nanocarriers and the need for novel options. The Introduction is currently quite confusing and the authors tend to use extracellular vesicles and exosomes interchangeably. The Introduction would read better if these are properly explained in different sections, one for introduction of extracellular vesicles, categories (include a figure?), source, purification methods etc. Some background on liposomes and hybrid nanocarriers would also be useful as well.

Response: Thank you for your comment. To address your comment, we have revised the introduction to clearly differentiate between exosomes and other extracellular vesicles. We have also included a new table to provide a detailed explanation of the categorization of extracellular vesicles. Additionally, the introduction has been rewritten for better clarity and now includes more detailed explanations about liposomes and the overall study design. We hope this updated version is more comprehensible. (Changes are highlighted in green.)

2. The methods section needs a lot more detail. 

Response: Thanks for your comment. To address your comment, we have enhanced the methods section by incorporating more detailed information and revising certain parts for better clarity. Please refer to the revised version, where the changes are highlighted in green.

- In Section 2.2: Were the BM-MSC patient-derived (human) or mouse?

Response: Thanks for the comment. In this study, we extracted exosomes from bone marrow-derived human mesenchymal stem cells (BM-hMSCs). Throughout the text, the term "BM-MSCs" has been replaced with "BM-hMSCs" to accurately reflect the cell source.

- Section 2.2.1: How was supernatant concentrated down 40-fold? Was the total volume 40mls? This is not very clear from the written protocol.

Response: Thanks for the comment. 50 ml of the BM-hMSCs supernatant was centrifuged (500 × g at 4 °C for 5 mins and then at 3000 × g at 4 °C for 10 mins) to remove dead cells and cell debris. The resulting supernatant was filtered through a 0.22 µm filter and concentrated down 40-fold using a 100 kDa filter (Corning Spin-X UF centrifugal concentrator, 100000 MWCO Membrane) through repeated centrifugation at 3000 × g at 4 °C for 25 mins. To address your comment, the text was updated for more clarity (highlighted in green).

- Section 2.2.2 and 2.2.3: What volume of PBS was the E- and P-Exosome pellets dissolved in?

Response: Thanks for the comment . The pellets were dissolved in 1 ml PBS solution. To address your comment, this information is now added to these sections.

- Section 2.3.1: Why was fraction 6-10 chosen to fix with 2.5 % glutaraldehyde. Please elaborate in the text.

Response: Thanks for the comment. As it is now clerarluy expresd in the revised version , in this study, we assessed the purity of the separated fractions by checking for the absence of proteins and confirming the presence of CD90 and CD63, which are characteristic markers of exosomes. Our results indicated that fractions 6-10 met the criteria for both purity and the presence of exosome markers, leading us to select these fractions for further analysis. This information is thoroughly discussed in the discussion section. To address your comment and to enhance clarity for the authors, we have added a note in parentheses in Method Section 2.3.1 (highlighted in green): For SEM sample preparation, purified exosomes from SEC method (fractions 6-10, which met the criteria for purified exosomes, were combined as a single sample) were initially fixed in a 2.5% glutaraldehyde solution in PBS for 10 mins.

- In Section 2.3.2, how was the EV diluted with PBS? What dilution factor was used?

Response: Thanks for the comment. The purified exosomes were diluted 1:2 with filtered (0.22 µm) PBS. The explanation was included in the text. 

- Section 2.4: What plasmid (pCas9-GFP) amount and Ca2+ amount was used? What is the neutral gas that was used here?

Response: Thanks for the comment. The plasmid (15 μg) and calcium ions (50 mM) were added to the lipids (285 μg). Nitrogen gas was used as a neutral gas during the preparation process. These details have been added to the text. 

- It would be good to include the agarose gel electrophoresis data showing plasmid integrity as a supplementary figure.

Response: Thanks for your comment. To address this comment, the image of agarose gel electrophoresis has been included in the Figure S2. 

- Section 2.7: Was a titration of the plasmid amount performed? Why was 15µg chosen? What concentration/amount of PEI was used?

Response: Thanks for your comment. The Transfection protocol acourding to the following refrences were 1:3 ratio, while highr conceterations of PEI has ben showed to have cytotoxicity.

we have now revised the text in section 2.7 “

The stock solution prepared for 5 wells contained 15 μg of plasmid, 45 μl of PEI and 1 ml of medium. The final concentration of PEI in each well was 1 μg/μl. 

- Please keep the time and volume units consistent throughout.

Response: Thanks for your comment. The units were checked and edited for consistency throughout the text. 

3. Results

- Please reference the figure number/sub-division while describing the result in the text, specifically in sections 3.1, 3.3 and 3.4

Response: Thanks for your comment. The figure numbers were edited in the revised text. 

- Section 3.1 and Section 3.5 can be combined into one figure.

Response: Thanks for the comment. To address your comment, two sections and the mentioned figures were combined as section 3.1 and figure 1. 

- Section 3.6: How was the trend analyzed? “Trend is evident” is a very vague statement.

Response: Thanks for the observation. We have rephrased the mentioned sentence. There is no “trend is evident” in the revised sentence. 

- Was a titration performed with both variable EV and plasmid concentrations? If so, this data should be included.

Response: Experiments varying the concentration of the EV also exhibited the proportionally higher concentration of plasmids were performed.

- The authors are encouraged to test other transfection agents in comparison as a control? To compare and assess the effectiveness of using mesenchymal stem-cell derived EVs as a nanocarrier.

Response: Thank you for your thoughtful suggestions. Although we cannot incorporate the additional analyses in the current study due to constraints, we find your recommendations valuable for consideration in future research.

- The authors are also encouraged to demonstrate the effectiveness of the EV nanocarrier in other cell types including cancer cells, T/B cells or any primary cell type. HEK293T cells is one of the easiest to transfect cell type. So, it would be powerful to demonstrate the effectiveness of EVs in alternate cell types.

Response: Thank you for your thoughtful suggestions. Although we cannot incorporate the additional analyses in the current study due to constraints, we find your recommendations valuable for consideration in future research.

4. - Section 4: In the first line, the authors say that MSCs were used as the source of EVs as high amounts were extracted compared to other cell types. How was this determined? Were other cell types tested? If so, it would be great to include that data.

Response: Thanks for your comment. The statement in the text regarding the relative exosome production of different cell types is based on findings reported in the published literature. For example, one study has shown that MSCs can produce higher amounts of exosomes compared to other cell lines, such as myoblasts, the human acute monocytic leukemia cell line (THP-1), and the human embryonic kidney cell line (HEK). In the revised version, to address your comment, the revised text cites the relevant published articles that support these comparative observations on exosome secretion levels across different cell types (1). 

Plese also note refrence 43 in the revised manuscript :

Wee R, Yeo Y, Chai R, Zhang B, Sim S, Yin Y, et al. Mesenchymal stem cell : An ef fi cient mass producer of exosomes for drug delivery ☆. Adv Drug Deliv Rev. 2013;65: 336–341. doi:10.1016/j.addr.2012.07.001

- What does ‘lowest protein contamination’ mean? How was this determined?

Response: Thanks for your comment. Protein level was quantified based on Bradford assay. This result indicated that there was no protein contaminant in fractions 6-10 (Figure 1a). Additionally, measuring OD280 confirmed the absence of protein contamination in this specific fractions. However the flowcytometry data showed expression of exsomme markers CD63, CD9 and CD90. 

- What is the advantage of using CL-4B over other sepharoses? Describe and please provide a reference.

Response: Thanks for your comment. The following text and related references were added to the discussion (highlighted in green) to address your comment. 

“In this study, sepharose CL-4B was chosen as the stationary phase due to its superior characteristics compared to other sepharoses like CL-2B and S-400, including a wider range of pH stability, the ability to withstand denaturants without compromising performance, and its suitability for separating a diverse range of biomolecules based on their size and shape. Sepharose CL-4B has an average pore size of 40 nm in comparison with CL-2B with 75nm pore size.” Please see discussion section 

- In this manuscript, no titration was shown for EV or exosome-hybrids, to determine transfection efficiency. How can this be translation to in vivo or clinical studies if this information is not available in this manuscript.

Response: The data presented in Figure 3 provide insights into the transfection efficiency of different formulations and concentrations. Comparing exosomes with the exosome-liposome hybrids showed that pure EVs (Sample 5) achieved greater transfection efficiency compared to exosome-liposome hybrids (Samples 1-4). This suggests that EVs can effectively deliver genetic material compared to non-toxic liposome prepared from soybean phosphatidylcholine (PC), CHOL, glycerol, and vitamin E (the Mozafari method).

Further optimization were performed by Increasing exosome concentrations: the results indicated that varying concentrations of EVs have a significant impact on transfection efficiency. Specifically, higher concentrations of EVs (0.1 ml and 0.2 ml) led to a decrease in transfection efficiency (Figure 3e-h, 3B). While the observed differences in transfection efficiency between EVs and exosome-l

---

## [Decision Letter · Decision Letter 1]

22 Nov 2024

Mesenchymal Stem Cell-Derived Exosome and Liposome Hybrids as Transfection Nanocarriers of Cas9-GFP Plasmid to HEK293T Cells

PONE-D-24-15406R1

Dear Dr. Sefidbakht,

We’re pleased to inform you that your manuscript has been judged scientifically suitable for publication and will be formally accepted for publication once it meets all outstanding technical requirements.

Kind regards,

Abdul Qadir Syed, PhD

Academic Editor

PLOS ONE

Additional Editor Comments (optional):

Reviewers' comments:

Reviewer's Responses to Questions

**Comments to the Author**

1. If the authors have adequately addressed your comments raised in a previous round of review and you feel that this manuscript is now acceptable for publication, you may indicate that here to bypass the “Comments to the Author” section, enter your conflict of interest statement in the “Confidential to Editor” section, and submit your "Accept" recommendation.

Reviewer #1: All comments have been addressed

2. Is the manuscript technically sound, and do the data support the conclusions?

Reviewer #1: Yes

3. Has the statistical analysis been performed appropriately and rigorously? 

Reviewer #1: Yes

4. Have the authors made all data underlying the findings in their manuscript fully available?

Reviewer #1: Yes

5. Is the manuscript presented in an intelligible fashion and written in standard English?

Reviewer #1: Yes

6. Review Comments to the Author

Reviewer #1: The authors have addressed all the comments satisfactorily and Manuscript can now be accepted in this current form.

7. PLOS authors have the option to publish the peer review history of their article (what does this mean?). If published, this will include your full peer review and any attached files.

Reviewer #1: No

---

## [Editor Report · Acceptance letter]

23 Dec 2024

PONE-D-24-15406R1 

PLOS ONE

Dear Dr. Sefidbakht, 

I'm pleased to inform you that your manuscript has been deemed suitable for publication in PLOS ONE. Congratulations! Your manuscript is now being handed over to our production team.

Kind regards, 

on behalf of

Dr. Abdul Qadir Syed 

Academic Editor

PLOS ONE